# Arf6 regulates the cycling and the readily releasable pool of synaptic vesicles at hippocampal synapse

Erica Tagliatti[1,2†], Manuela Fadda[2†], Antonio Falace[2‡], Fabio Benfenati[1,2], Anna Fassio[1,2]*

[1]Center of Synaptic Neuroscience and Technology, Istituto Italiano di Tecnologia, Genova, Italy; [2]Department of Experimental Medicine, University of Genova, Genova, Italy

**Abstract** Recycling of synaptic vesicles (SVs) is a fundamental step in the process of neurotransmission. Endocytosed SV can travel directly into the recycling pool or recycle through endosomes but little is known about the molecular actors regulating the switch between these SV recycling routes. ADP ribosylation factor 6 (Arf6) is a small GTPase known to participate in constitutive trafficking between plasma membrane and early endosomes. Here, we have morphologically and functionally investigated Arf6-silenced hippocampal synapses and found an activity dependent accumulation of synaptic endosome-like organelles and increased release-competent docked SVs. These features were phenocopied by pharmacological blockage of Arf6 activation. The data reveal an unexpected role for this small GTPase in reducing the size of the readily releasable pool of SVs and in channeling retrieved SVs toward direct recycling rather than endosomal sorting. We propose that Arf6 acts at the presynapse to define the fate of an endocytosed SV.

*For correspondence: afassio@unige.it

†These authors contributed equally to this work

Present address: ‡Institut national de la santé et de la recherche médicale, Institut de Neurobiologie de la Méditerranée, Marseille, France

Competing interests: The authors declare that no competing interests exist.

## Introduction

Synaptic vesicles (SVs) are continuously recycled at the nerve terminal and this process is tightly controlled to ensure the fidelity of neurotransmission. After endocytosis, SVs are shuttled back to the recycling pool either directly or via endosome-like compartments. The existence of endosome-like compartments has been demonstrated at the synapse since the pioneering experiments on synapse ultrastructure (*Heuser and Reese, 1973*), but their functional role in SV cycling is highly debated and the molecular and signaling pathways that define the fate of an endocytosed SV are still largely unknown (see for recent reviews *Morgan et al., 2013*; *Kokotos and Cousin, 2015*; *Jähne et al., 2015*).

The ADP ribosylation factor 6 (Arf6) is a small GTPase that localizes primarily to the plasma membrane/endosomal system and is best known as a regulator of endocytic trafficking and actin cytoskeleton dynamics (*D'Souza-Schorey and Chavrier, 2006*; *Myers and Casanova, 2008*). As for others small GTPases, the activity of Arf6 is tightly controlled by two families of regulatory proteins: guanine nucleotide exchange factors (GEFs) that function as Arf6 activators by facilitating the conversion from GDP-bound inactive form to GTP-bound active form, and GTPase activating protein (GAPs) that function as negative regulators by enhancing GTP hydrolisis. Neurons express several GEFs and GAPs for Arf6 and Arf6 has been reported to play a role in neurite outgrowth and dendritic spine maturation although its function in mature synapses is largely unknown (*Jaworski, 2007*). At postsynaptic sites, accumulating evidence has recently shown that Arf6 plays a role in AMPA receptor trafficking and long-term synaptic plasticity (*Scholz et al., 2010*; *Myers et al., 2012*; *Oku and*

**eLife digest** Communication between neurons takes place at cell-to-cell contacts called synapses. Each synapse is formed between one neuron that sends the message, and another neuron that receives it. The neuron before the synapse – called the presynaptic neuron – contains packets called synaptic vesicles, which are full of chemical messengers ready to be released upon activity.

Accurate communication between neurons relies on the exact composition, and organized trafficking, of the synaptic vesicles when the neuron is active. Synapses also contain bigger structures, called endosomal structures, which may represent an intermediate station in which synaptic vesicle composition is controlled. However, the trafficking of synaptic vesicles through the endosomal structures is poorly understood.

Now, Tagliatti, Fadda et al. have revealed that a protein called Arf6 plays an important role in presynaptic neurons. The experiments involved rat neurons grown in the laboratory, and showed that Arf6 controls both the number of synaptic vesicles ready to be released and the trafficking of synaptic vesicles via endosomal structures in active neurons. The next step following on from these findings is to understand how Arf6 exerts its effects and how this protein is regulated in the presynaptic neuron.

*Huganir, 2013*; *Zheng et al., 2015*). At presynaptic sites, Arf6 has been predicted to play a role in the assembly of the clathrin coated complex during SV endocytosis (*Krauss et al., 2003*). While overexpression of the Arf6 GEF msec7-1 has been reported to increase basal synaptic transmission at the *Xenopus* neuromuscular junction (*Ashery et al., 1999*), the function of Arf6 at the presynaptic terminal has never been directly addressed. Interestingly, mutations in Arf6 regulatory genes have been recently associated with intellectual disability and epilepsy in humans (*Shoubridge et al., 2010*; *Falace et al., 2010*; *Rauch et al., 2012*; *Fine et al., 2015*).

Here, we investigate the ultrastructural and functional effects of Arf6 silencing in hippocampal synapses and reveal an unexpected presynaptic role for this small GTPase in determining the size of the readily releasable pool of SVs and in promoting direct *versus* endosomal recycling of SVs.

## Results

We first investigated on the expression of the small GTPase Arf6 at synaptic level by biochemical experiments and revealed expression of Arf6 in isolated nerve terminal-extract; differential extraction of synaptosomal proteins (*Phillips et al., 2001*) revealed that Arf6 is not tightly associated with presynaptic or postsynaptic membranes, as it is mainly extracted at pH6 similarly to the SV protein synaptophysin. We also evaluated Arf6 expression at synaptic level by immunocytochemistry. Endogenous Arf6 colocalyzed with both presynaptic (Synaptophysin) and postsynaptic (Homer1) markers in primary rat hippocampal neurons (17 days in vitro, DIV) and triple labelling showed expression of the small GTPase at the presynaptic and postsynaptic site of single synaptic puncta (*Figure 1—figure supplement 1*). To directly examine how Arf6 activity impacts on synapse structure, we performed electron microscopy (EM) analysis at Arf6-knockdown (KD) synapses. Rat hippocampal neurons were transduced at 12 DIV, after the initial wave of synaptogenesis had occurred, with a lentiviral vector, driving the expression of short hairpin targeting the coding sequence of the rat Arf6 mRNA (shRNA#1) or the respective mismatch control and GFP as a reporter. The silencing efficiency was tested 5 days post transduction by western blotting (WB) and immunocytochemistry (ICC) (*Figure 1—figure supplement 2*). Ultrastructural analysis revealed that Arf6-KD synapses were undistinguishable from control synapses in terms of synaptic area and active zone (AZ) length (*Supplementary file 1*), but were characterized by a decreased total number of SVs and a significantly increased number of SVs docked at the AZ (*Figure 1A*). Moreover, intraterminal cisternae, resembling endosome-like structures and occasionally found in control synapses, were dramatically increased in Arf6-silenced synapses (*Figure 1A*). The observed phenotype was completely rescued by the expression of a rat Arf6 variant resistant to shRNA#1 silencing (Arf6-res, *Figure 1—figure supplement 3*).

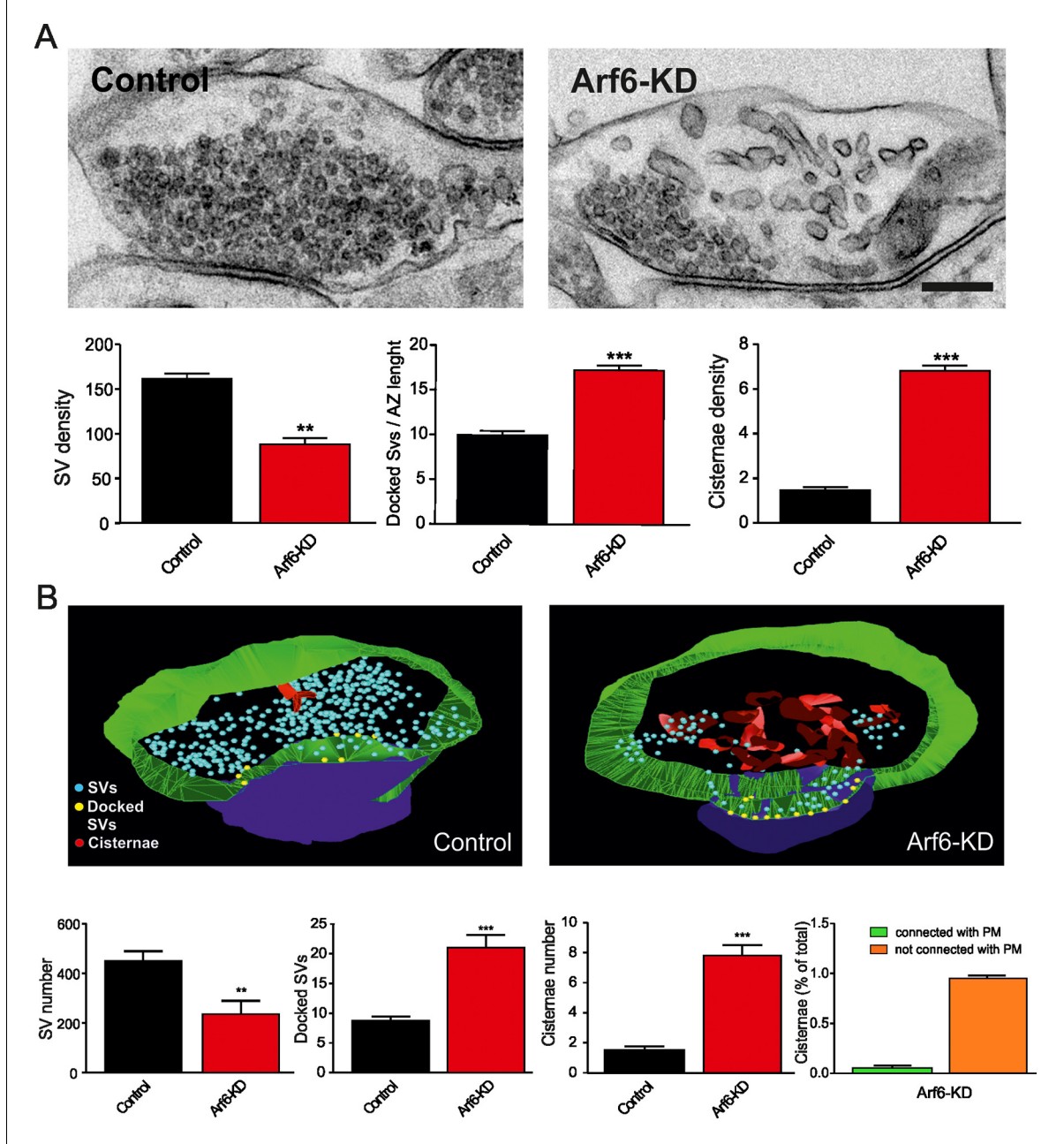

**Figure 1.** Reduced SV density and accumulation of intraterminal cisternae at Arf6 deficient synapses. (**A**) *Upper panels*, representative electron micrographs of synaptic terminals from cultured hippocampal neurons (17 DIV) transduced with lentiviruses expressing an Arf6 shRNA (Arf6-KD) or an inactive mismatched version (Control). Scale bar, 200 nm. *Lower panels*, morphometric analysis of the density of SVs, docked SVs and cisternae in control (black) and Arf6-silenced (red) synapses. Data are means ± SEM from 3 independent preparations (n=134 and 162 synapses for control and Arf6-KD, respectively). (**B**) *Upper panels,* representative 3D synapse reconstructions from 60 nm-thick serial sections obtained from hippocampal neurons transduced as in A. Total SVs, docked SVs, presynaptic plasma membrane, postsynaptic density and cisternae are shown in light blue, yellow, green, blue and red respectively. *Lower panels*, morphometric analysis of number of SVs, docked SVs and cisternae in control (black) and Arf6-KD (red) synapses. Right panel shows the percentage of intraterminal cisternae that are connected (green) or not-connected (orange) with the plasma membrane for Arf6-KD synapses. Data are means ± SEM from 10 (Control) and 10 (Arf6-KD) reconstructed synapses. Statistical analysis was performed with the unpaired Student's *t*-test. **p<0.005, ***p<0.001, versus control.

The following figure supplements are available for figure 1:

**Figure supplement 1.** Synaptic localization of Arf6.

*Figure 1 continued*

**Figure supplement 2.** Tools for silencing and rescue Arf6 expression.

**Figure supplement 3.** Rescue of the Arf6-KD ultrastructural phenotype.

Three-dimensional (3D) reconstruction of Arf6-silenced and control synapses confirmed the decreased SV density, the increased number of docked SVs and intraterminal cisternae and revealed that most if not all of the latter structures were separated from the plasma membrane (*Figure 1B*). This observation rules out the possibility that the intraterminal cisternae represent membrane infoldings in continuity with the plasma membrane. To investigate on the nature of intraterminal cisternae, Arf6-silenced and control neurons were stained with the synaptic marker VAMP2 and the synaptic endosome markers Rab5 and Vti1A. At Arf6-KD VAMP2-positive synapses, a significant increase of both Rab5 and Vti1A signals was observed (*Figure 2*). The increased immmunoreactivity for endosomal markers at synaptic sites together with the direct observation that cisternae that accumulate upon Arf6 silencing are detached from the plasma membrane lead us to consider them *bona fide* endosome-like organelles (Elos).

To assess whether the morphological phenotype was due to the loss of Arf6 protein per se or rather to its impaired activation we tested the effect of SecinH3, an inhibitor of the GEF of the cytohesin family (*Hafner et al., 2006*) that prevents the activation of Arf proteins. We first tested the efficiency of SecinH3 to impair Arf6 activation in hippocampal primary cultures (17 DIV) by pull-down

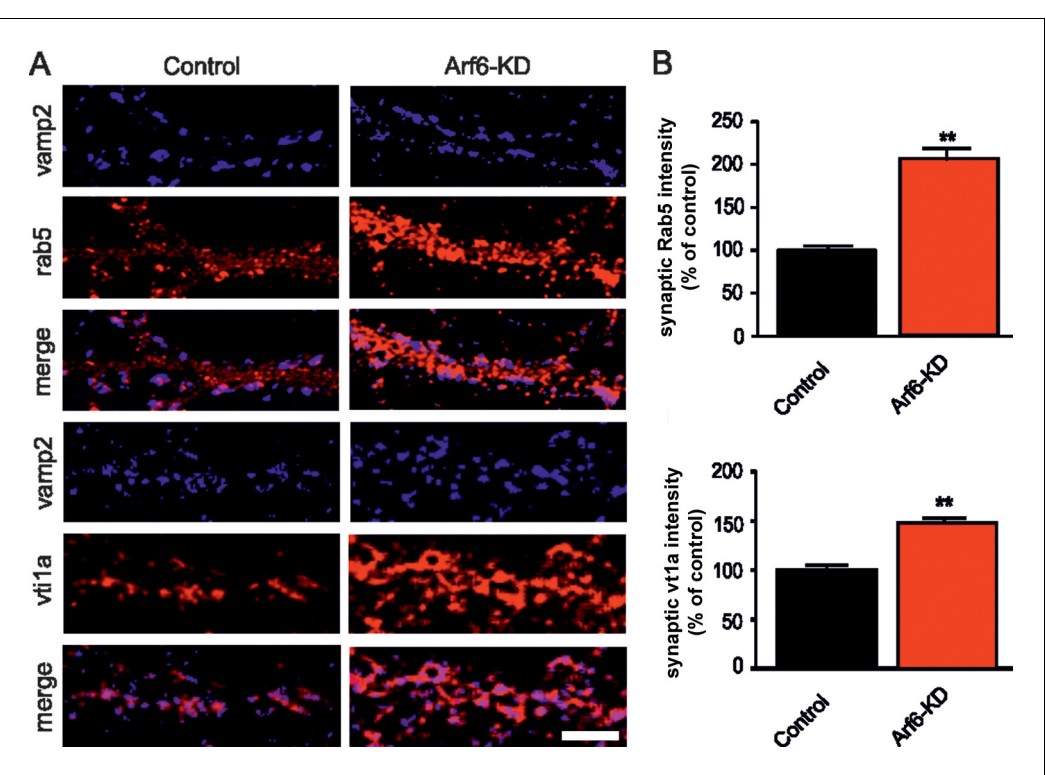

**Figure 2.** Increased expression of endosomal markers at Arf6-deficient synapses. (**A**) Representative images of synapses from rat hippocampal neurons (17 DIV) transduced with either Arf6 shRNA (Arf6-KD) or an inactive mismatched version (Control) and immunostained with anti-Vamp2 (blue) and either anti-Rab5 or anti-Vti1A (red) antibodies. Scale bar, 5 µm. (**B**) Intensity values for Rab5 and Vti1A signal at VAMP2-positive puncta in control (black) and Arf6-silenced (red) synapses. Data are means ± SEM from 3 independent preparations. 500 synapses have been counted for each preparation. Statistical analysis was performed with the unpaired Student's *t*-test. **p<0.005 versus control.

experiments and revealed a significant decrease of GTP-bound Arf6 in SecinH3-treated neurons with respect to control (*Figure 3—figure supplement 1*). Wild-type neurons were therefore treated with either SecinH3 or vehicle (DMSO) and analyzed for synapse morphology. SecinH3 treatment did not result in any variations in the synaptic area or AZ length (*Supplementary file 1*), but the complex ultrastructural modifications observed at Arf6-silenced synapses (i.e. decreased number of SVs, increased number of docked SVs and increased number of intraterminal cisternae) were completely phenocopied (*Figure 3A*). The possibility that the SecinH3 treatment results in delocalization of Arf6 from synaptic sites was assessed by staining control and treated cultures with the presynaptic marker Synaptophysin and Arf6. No significant difference was observed in the intensity of the Arf6 signal at Synaptophysin-positive puncta in control and SecinH3-treated neurons (*Figure 3B*). These data demonstrate that the lack of Arf6 activation, rather than of its expression, is sufficient to cause the accumulation of both SVs at the AZ and Elos at the expense of total SVs (*Figure 3*). Moreover, the acute SecinH3 treatment suggests that accumulating Elos represent transient structures rather than stable organelles recruited at the synapse upon loss of Arf6 activation.

To test this hypothesis we challenged control and Arf6-KD neurons with tetrodotoxin (TTX) to block spontaneous activity in the network and the related SV trafficking. As expected TTX had no effect on control synapses, because the performed treatments were too short to induce homeostatic changes (*Turrigiano, 2012*; *Verstegen et al., 2014*). In contrast, the blockade of spontaneous activity with 1 µM TTX for either 12 or 24 hrs, completely reverted the increase in intraterminal Elos and the decrease in SV density observed upon Arf6 silencing (*Figure 4A,B*). TTX treatments did not result in any variations in the synaptic area or AZ length in both experimental groups (*Supplementary file 1*). These data unequivocally demonstrate that, at Arf6-KD synapses, intraterminal Elos represent an activity-dependent intermediate station in the SV recycling pathway.

Interestingly, the increased number of docked SVs at Arf6-KD synapses, was unaffected by TTX treatment (*Figure 4A,B*), suggesting a distinct mechanism by which Arf6 activation regulates the abundance of docked SVs, or a differential sensitivity to network activity.

To couple the synaptic ultrastructure with functional analysis of SV trafficking, we employed SynaptophysinpHluorin (Syphy), a fluorescent probe exquisitely designed to monitor the SV exo-endocytosis cycle at single synapses (*Miesenböck et al., 1998*; *Fassio et al., 2011*). Arf6-shRNA (or the mismatched control) together with the RFP reporter, and SypHy were transiently co-transfected in rat hippocampal neurons in order to record from a silenced neuron embedded in a network of non-silenced ones (*Figure 5—figure supplement 1*, *Figure 5—figure supplement 2*). Considering the emerging role of Arf6 in AMPA receptor trafficking (*Scholz et al., 2010*; *Myers et al., 2012*; *Oku and Huganir, 2013*; *Zheng et al., 2015*), this approach allows ruling out the contribution of postsynaptic effects on the observed phenotype. Experiments were performed at 17–18 DIV (3–4 days after transfection) and RFP/SypHy-positive axonal processes were selected for stimulation and analysis (*Figure 5A*). When the amount of surface-expressed SypHy was evaluated, we observed a significant increase of extracellularly exposed SypHy at Arf6-silenced synapses compared with control synapses, suggestive of a global impairment of SV cycling (*Figure 5—figure supplement 2*). Neurons were first stimulated with 40 action potentials (APs) at 20 Hz, a protocol widely employed to measure the readily releasable pool (RRP) of SVs and, therefore, suitable to reveal if the increased docked SVs observed by EM were functional or represented mistargeted/unprimed SVs, not competent for fusion. Both the shRNA #1 (*Figure 5B,C*) and the shRNA #2 (*Figure 5—figure supplement 2*) significantly increased the peak fluorescence at the end of the 2 s. Stimulation without affecting the kinetics of fluorescence return to basal level at the end of the stimulation, which describe 'post-stimulus' endocytosis. To reveal whether the increased peak fluorescence was due to higher exocytosis or impairment of 'during-stimulus' endocytosis we employed the $H^+$ATPase inhibitor Bafilomycin. This drug prevents reacidification of internalized SVs and allows monitoring net exocytosis. The peak fluorescence was equally increased when neurons were stimulated in the presence of Bafilomycin, suggesting that the observed peak fluorescence increase was indeed attributable to an increased number of exocytosed SVs. (*Figure 5D*). Next, to exclude the involvement of shRNA-mediated off-target effects, we performed rescue experiments. When hippocampal neurons were triple transfected with SypHy, Arf6 shRNA#1 and a V5-tagged Arf6 rat variant resistant to shRNA#1 silencing (Arf6-res, *Figure 5—figure supplement 1*) the peak fluorescence at the end of the 40APs@20Hz stimulation returned to the control level (*Figure 5E*). These data demonstrate that increased docked SVs in Arf6-depleted synapses are indeed functional releasable SVs and that Arf6

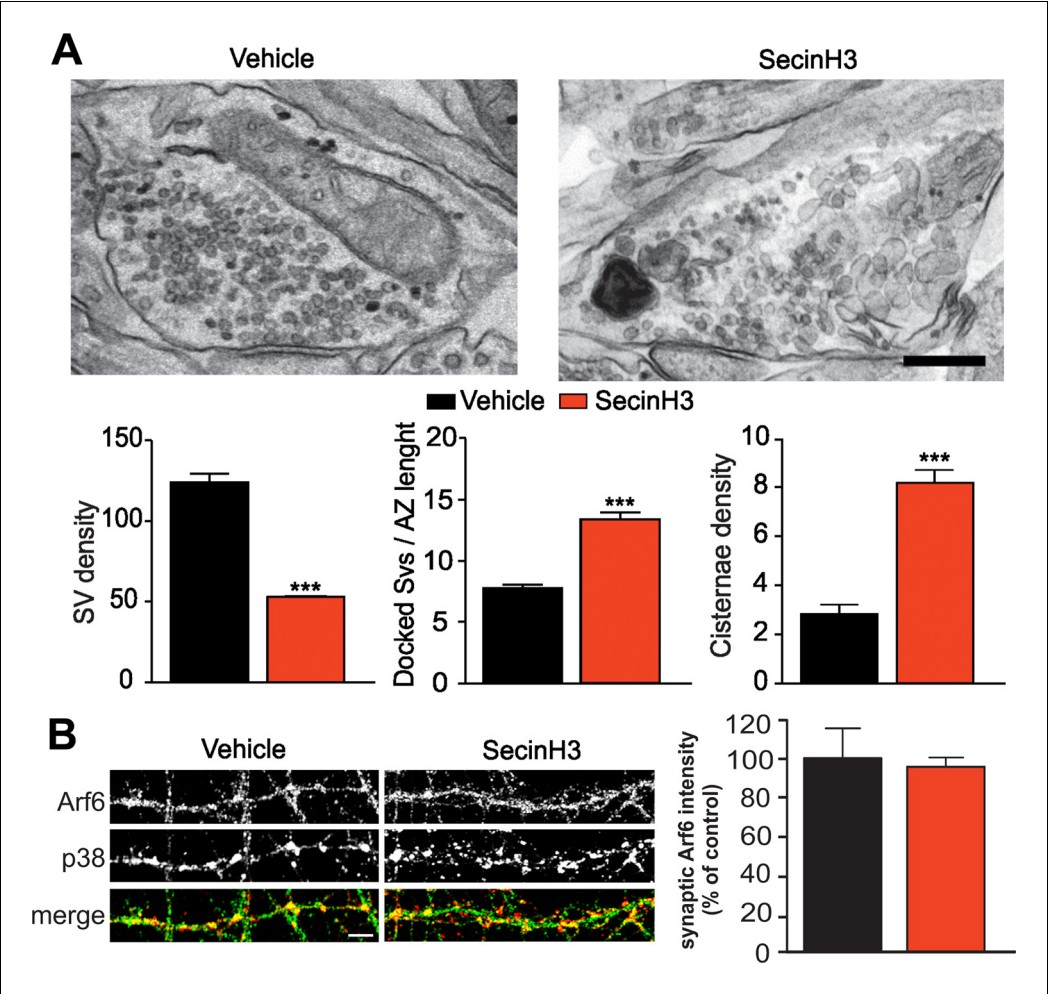

**Figure 3.** SecinH3 treatment phenocopies Arf6 silencing. (**A**) *Upper panels*, representative electron micrographs of synaptic terminals from cultured hippocampal neurons (17 DIV) treated with either SecinH3 (30 µM) or DMSO (Vehicle). *Lower panels*, morphometric analysis of the density of SVs, docked SVs and cisternae in control (black) and SecinH3 treated (red) synapses. Data are means ± SEM from 3 independent preparations (n=75 for experimental group). Statistical analysis was performed with the unpaired Student's *t*-test. \*\*\*p<0.001, versus respective control. Scale bar, 200 nm. (**B**) *Left panels*, representative images of synapses from rat hippocampal neurons (17 DIV) treated with SecinH3 (30 µM) or DMSO (Vehicle) and immunostained with anti-Arf6 and anti-synaptophysin (p38) antibodies. Scale bar, 5 µm. *Right panel*, intensity values for Arf6 signal at p38-positive puncta in control (black) and SecinH3-treated (red) synapses. Data are means ± SEM from 3 independent preparations. 300 synapses have been counted for each preparation.

The following figure supplement is available for figure 3:

**Figure supplement 1.** SecinH3 treatment inhibits Arf6 activation in hippocampal primary neurons.

silencing causes an alteration in the process of neurotransmission in the absence of any effects on the kinetics of RRP endocytosis.

Then, to reveal if accumulation of intraterminal Elos also impacts on neurotransmission, a longer stimulation protocol (400APs@20Hz), which allows to recruit and recycle a larger SV pool, have been employed. Bafilomycin was concomitantly used to dissect out 'during-stimulus' recycling. At control synapses, Bafilomycin increased the response to the stimulation, indicative of the existence of an active 'during-stimulus' recycling (*Figure 6A*). At the end of the stimulation, and in the absence of Bafilomycin, fluorescence returned to the basal level due to 'post-stimulus' endocytosis, with an average time constant of endocytosis (τ) of 36 s. Strikingly, at Arf6-KD synapses, the peak

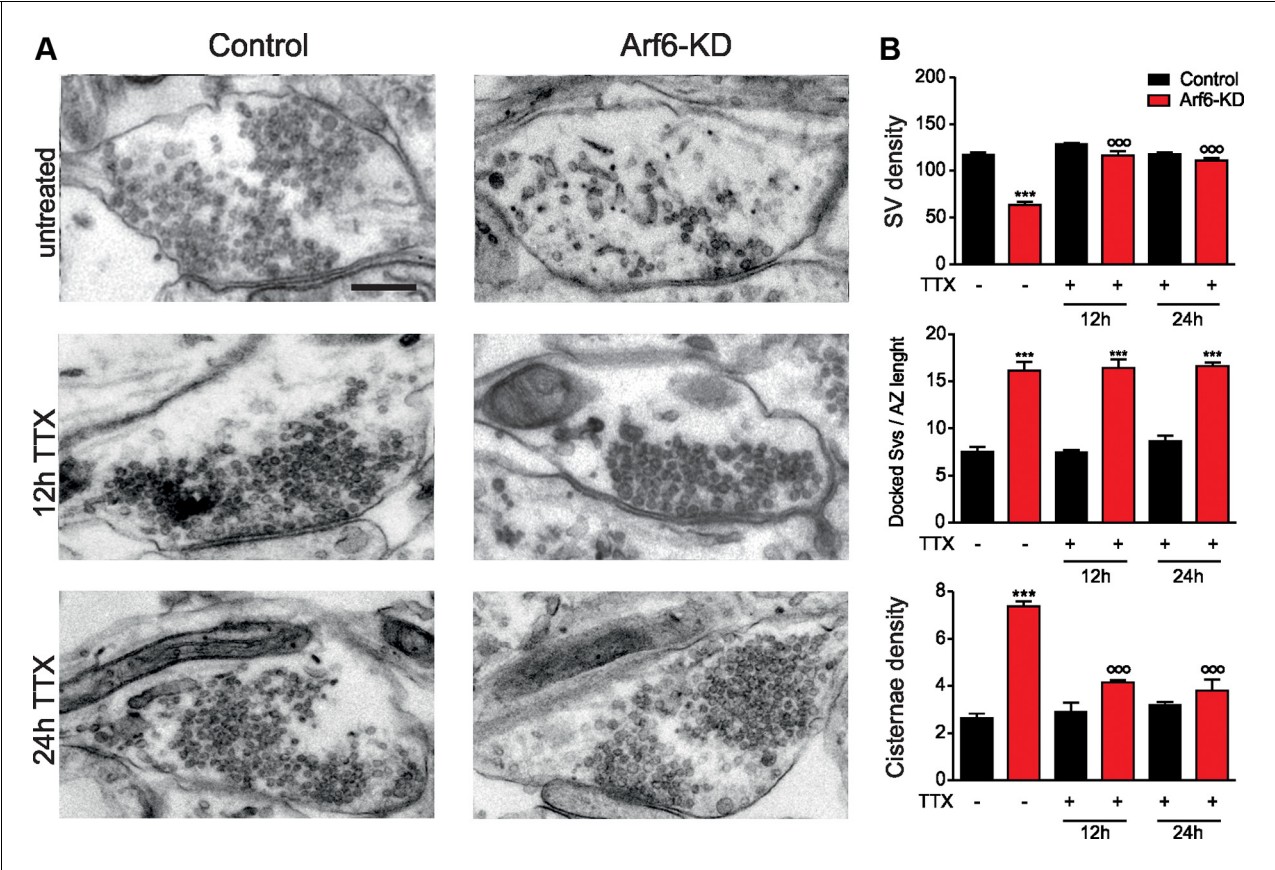

**Figure 4.** The formation of intraterminal cisternae in Arf6-deficient synapses is activity dependent. (**A**) Representative electron micrographs of synaptic terminals from cultured hippocampal neurons (17 DIV) transduced with Arf6-KD or an inactive mismatched version (Control). Neurons were either left untreated or treated with 1 µM TTX for 12 or 24 hr. Scale bar, 200 nm. (**B**) Morphometric analysis of the density of SVs, docked SVs and cisternae in control (black) and Arf6-KD (red) synapses. Data (80 synapses/group) are means ± SEM from 3 independent preparations. Statistical analysis was performed with two-way ANOVA followed by the Bonferroni's multiple comparison test. ***$p<0.001$ versus control; °°°$p<0.001$ versus the respective untreated sample.

fluorescence was not modified by Bafilomycin, suggesting a strong impairment of 'during-stimulus' endocytosis (*Figure 6A*). To measure the net exocytosis (EXO) during stimulation, the areas below the curves in the presence of Bafilomycin were calculated and no differences were observed between control and Arf6 KD synapses (*Figure 6B*). To quantify net 'during-stimulus' endocytosis (ENDO), additional curves were obtained by plotting the point-by-point differences between the curves run in the presence and absence of Bafilomycin. The time course of 'during- stimulus' endocytosis revealed that recycling was virtually absent in the first 12 s of stimulation, and started to take place only in the last 8 s of stimulation, resulting in a significant decrease in ENDO upon Arf6 silencing (*Figure 6B*). Moreover, Arf6 silencing slowed down also 'post-stimulus' endocytosis as revealed by the significantly higher τ (59 s, *Figure 6A*), suggesting an involvement of the small GTPase in the endocytosis of recycling SVs, differently from what has been observed for SVs belonging to the RRP (*Figure 5C*). Remarkably, all the described defects on 'during-stimulus' and 'post-stimulus' recycling in Arf6-KD synapses were completely rescued by the concomitant expression of the shRNA#1-resistant Arf6 (*Figure 6A,B*). The functional data demonstrate that SV recycling is impaired at Arf6 depleted synapses and this impairment is paralleled by an accumulation of synaptic Elos.

We next asked whether the accumulation of synaptic Elos at Arf6 depleted synapses is attributable to a block of SV reformation from intermediate endosomal structures or to the fact that SVs are constrained to pass through intermediate endosomal structures instead of recycling directly. To challenge these hypotheses, we visualized SVs and intraterminal cisternae formation upon AP firing by stimulating control or Arf6-KD neurons (17–18 DIV) in the presence of soluble HRP. Samples were

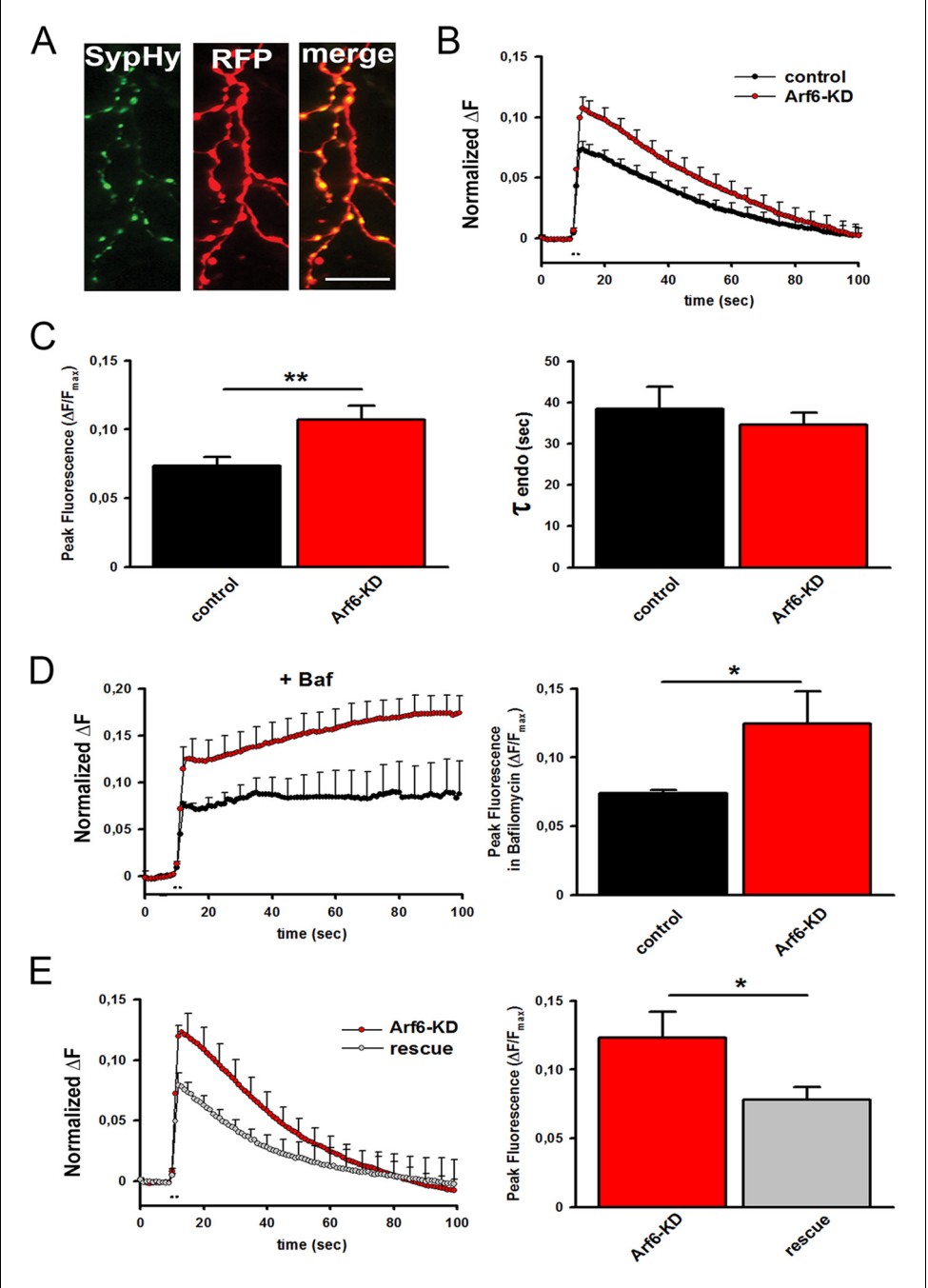

**Figure 5.** Increased readily releasable pool at Arf6-deficient synapses. (**A**) Representative images from neuronal processes cotransfected with Arf6-shRNA #1 (RFP) and SypHy. The merge panel shows colocalization at synaptic puncta. Scale bar, 10 μm. (**B**) Ensemble average traces of SypHy fluorescence plotted for control (n=18, black trace) and Arf6-KD (n=20, red trace) neurons stimulated with 40APs@20Hz (dotted line). Data are means with SEM values shown every 5 time points. (**C**) Means ± SEM of the peak fluorescence at the end of the stimulus (left) and of the time constant of endocytosis (τ endo, right), evaluated by fitting the fluorescence decay after stimulation by a single exponential function, for control (black) and ARF6-KD (red) synapses. (**D**) *Left panel*, ensemble average traces of SypHy fluorescence plotted for control (n=18, black trace) and Arf6-KD (n=20, red trace) neurons stimulated with 40APs@20Hz (dotted line) in the presence of 1 μM Bafilomycin (+Baf). Data are means with SEM values shown every 5 time points. *Right panel*, means ± SEM of the peak fluorescence at the end of the stimulus in the presence of 1 μM Baf. (**E**) *Left panel*, ensemble average traces of SypHy fluorescence plotted for Arf6-KD (n=10, red trace) and Arf6-KD neurons cotransfected with the Arf6 shRNA#1 resistant form (rescue, n=13, gray

*Figure 5 continued on next page*

*Figure 5 continued*

trace) and stimulated with 40APs@20Hz (dotted line). Data are means with SEM values shown every 5 time points. *Right panel*, mean ± SEM of peak fluorescence at the end of the stimulus in Arf6-KD (red) or rescued (gray) synapses. Statistical analysis was performed with the unpaired Student's *t*-test. *p<0.05; **p <0.01.

The following figure supplements are available for figure 5:

**Figure supplement 1.** Tools for silencing and rescue Arf6 expression by transfection.

**Figure supplement 2.** SypHy expression at Arf6-silenced synapses and effect of Arf6 shRNA#2.

fixed at the end of the stimulation (400APs@20Hz) and after 2 and 20 min. wash in the absence of HRP. The decreased number of SVs and the increased number of docked SVs and intraterminal cisternae were fully confirmed in Arf6 deficient synapses at all time point (*Figure 7—figure supplement 1*). However, when the HRP-positive (HRP$^+$) structures at the end of the stimulus, representative of active cycling during stimulation, were analyzed, a significant decrease HRP$^+$ SVs accompanied by an increase of HRP$^+$ cisternae in Arf6-KD terminals compared to control was observed (*Figure 7*). This result correlates with the observed defect in 'during-stimulus' endocytosis revealed by the SypHy experiments (*Figure 6*) and suggests that Elos preferentially form when Arf6 signalling is occluded. Interestingly, after 2 min. wash in the absence of HRP, HRP$^+$ SVs remained substantially unchanged in control synapses, while they significantly increased in Arf6-depleted terminals (*Figure 7*). This increase demonstrates that, in Arf6-depleted terminals, SVs efficiently form from intermediate endosomal structures, since the additional HRP$^+$SVs can only derive from intraterminal HRP$^+$-Elos after HRP had been removed from the extracellular medium. The data suggest that Arf6 regulates the direct formation and recycling of SVs so that, in its absence, Elos form upon stimulation and SVs lately bud from the newly formed structures. Finally, after 20 min-wash in the absence of HRP, SVs and intraterminal cisternae significantly lose their HRP content in both experimental groups, suggesting that both SVs and cisternae are actively recycled, thus ruling out the possibility that SVs are stuck at Elos in Arf6-depleted synapses (*Figure 7*).

## Discussion

In the present study, we uncovered a crucial function for the small GTPase Arf6 in the maintenance of RRP and determining recycling route of endocytosed SVs at the presynaptic terminal.

Our morpho-functional analysis demonstrated that upon Arf6 depletion synaptic Elos preferentially form during activity at the expense of SVs. Considering that the molecular nature, as well as the function of synaptic endosomal structures is a matter of debate, our 3D analysis coupled with staining for endosomal markers demonstrate that the structures that form upon Arf6-silencing are *bona fide* synaptic Elos. Moreover, these organelles unequivocally participate in SV recycling as their formation is abrogated by TTX treatment. The same phenotype is observed when blocking Arf6 activation by pharmacological treatment, demonstrating that synaptic Elos form due to the loss of Arf6 activation, that results therefore essential for the direct recycling of endocytosed SV. The increased travelling of SVs via synaptic Elos at Arf6-depleted synapses, results in recycling defects during long-lasting stimulation (20 s), when multiple rounds of exo-endocytosis are required, suggesting that direct, rather than endosomal, recycling is the favorite and most efficient recycling route during repetitive stimulation. Moreover, synaptic Elos formation is accompanied by an increased RRP demonstrating that, while defining the recycling route of endocytosed SV, Arf6 also regulates the abundance of release competent SVs at the AZ.

The function of endosomal structures at the synaptic terminal is still elusive, but a role in both regeneration of SVs and SV protein sorting and renewal has been described (*Hoopmann et al., 2010*; *Watanabe et al., 2014*; *Wucherpfennig et al., 2003*; *Uytterhoeven et al., 2011*; *Fernandes et al., 2014*). Our data reveal that the small GTPase Arf6 is a component of the machinery that regulates the fate of an endocytosed SV at the synapse. We speculate that active Arf6 shifts SVs toward direct recycling, away from synaptic Elos, through which only a subset of SVs, either devoid of Arf6 or in which Arf6 is inactive, travel to undergo protein turnover. This scenario is

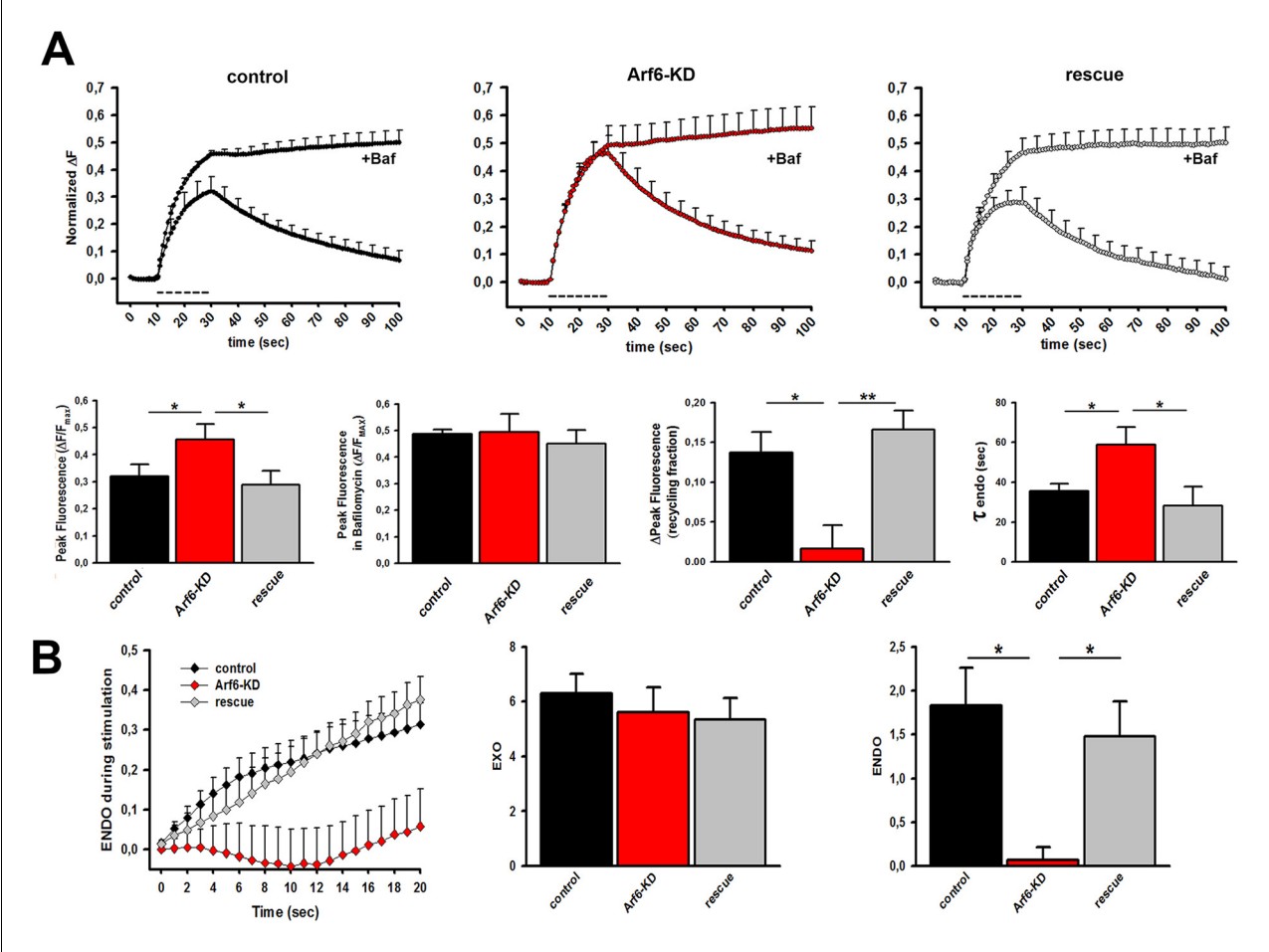

**Figure 6.** Decreased SV recycling at Arf6-deficient synapses. (**A**) *Upper panels*, ensemble average traces of SypHy fluorescence plotted for control (n=9, left), Arf6-KD (n=8, center) and rescued (n=8, right) neurons stimulated with 400APs@20Hz (dotted line) in the absence or presence of 1 μM Bafilomycin (+ Baf.). Data are means with SEM values shown every 5 time points. *Lower panels (from left to right)*, peak fluorescence at the end of the stimulus in the absence of Baf.; peak fluorescence at the end of the stimulus in the presence of Baf.; 'during-stimulus' recycling fraction calculated as the difference between the peak fluorescence in the presence and absence of Baf.; time-constants of 'post-stimulus' endocytosis evaluated by fitting the fluorescence decay after stimulation in the absence of Baf. by a single exponential function. Dara are means ± SEM. (**B**) *Left panel*, Time courses of endocytosis during stimulation with 400 APs at 20 Hz in control (black), Arf6-KD (red) and rescued (gray) synapses for the same experiments shown in A. Time courses were derived by subtracting the fluorescence trace in the presence of Bafilomycin (ΔFexo) from the trace in its absence (ΔFexo-ΔFendo). *Middle and right panels*, Rate of exocytosis (EXO) and endocytosis (ENDO) during stimulation were calculated as the area under the curves. Statistical analysis was performed with one-way ANOVA followed by the Bonferroni's multiple comparison test. *p<0.05, **p<0.01 versus Arf6-KD.

reminiscent of the role of active Arf6 in endosomal recycling of transferrin receptors, where the small GTPase mediates fast recycling of the receptor, and confirms the central role of Arf6 in determining the intracellular trafficking pathway of endocytic cargoes (*Montagnac et al., 2011*; *Grant and Donaldson, 2009*). Intriguingly, upstream regulators of Arf6 activity have been recently demonstrated to act in SV recycling at the *Drosophila* neuromuscular junction with morphological and functional phenotypes reminiscent of the Arf6-depleted hippocampal synapses described in this paper (*Uytterhoeven et al., 2011*; *Podufall et al., 2014*). Several downstream effectors for active GTP-bound Arf6 have been described that could play a role in SV recycling at the plasma membrane (*Donaldson and Jackson, 2011*), and among them the adaptor protein AP-2 is of particular relevance (*Paleotti et al., 2005*). Neuron-specific deletion of AP-2 has been recently described to result in the accumulation of presynaptic endosome-like vacuoles and loss of SVs at hippocampal synapses (*Kononenko et al., 2014*), resembling the Arf6-KD phenotype described here. Accumulation of synaptic endosomal structures can be attributed to defective SVs reformation or to an increase in the

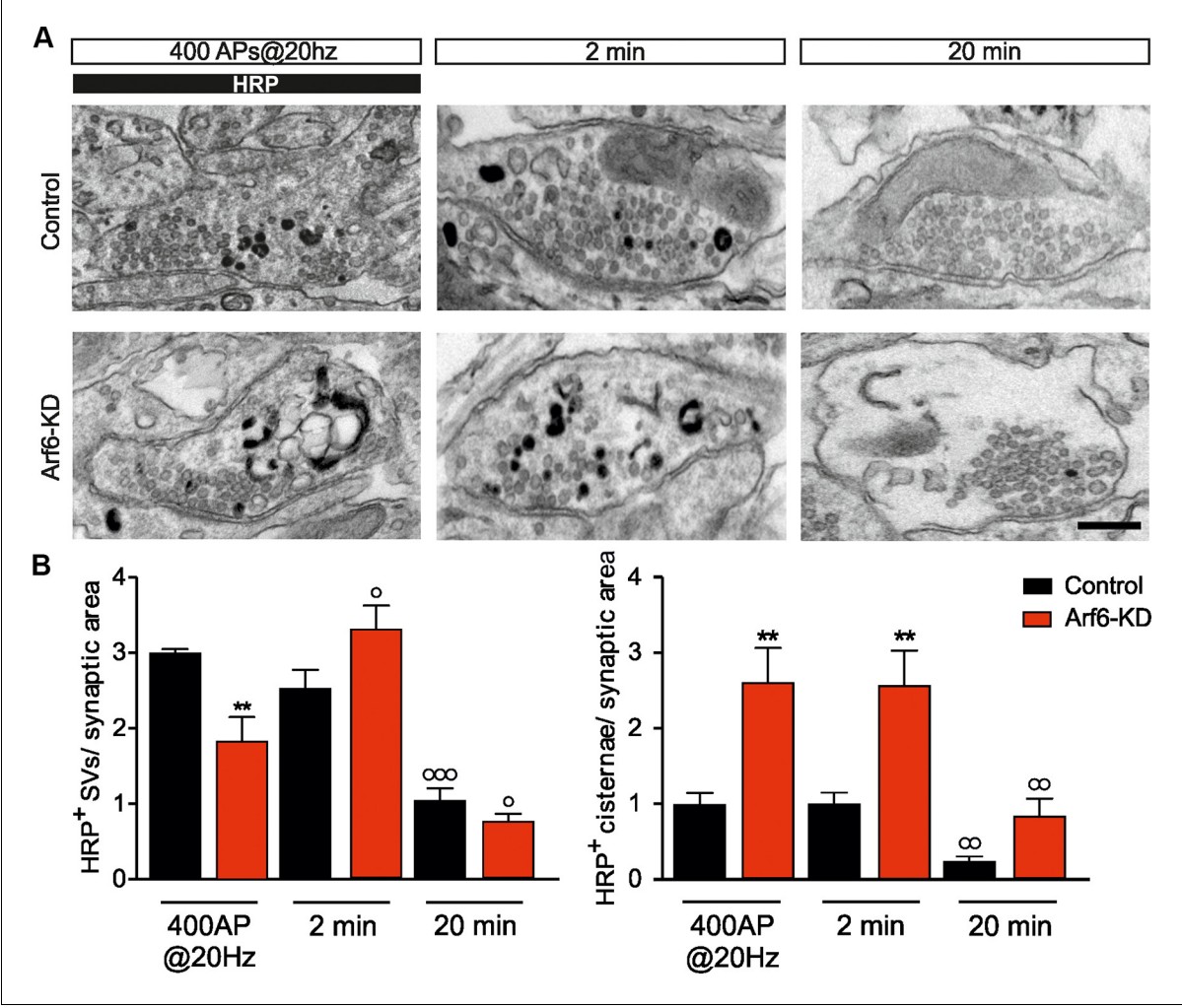

**Figure 7.** Stimulus-induced formation of intraterminal cisternae is increased at the expense of SV formation at Arf6-silenced synapses. (**A**) Representative EM images for Control and Arf6-KD synapses in neurons stimulated with 400APS@20hz in the presence of HRP (10 mg/ml) and then incubated for additional 2 or 20 min, in the absence of HRP. Scale bar, 200 nM. (**B**) Quantitative analysis of SV and cisternae dynamics measured by HRP labelling in control and Arf6-KD synapses treated as in A. Data (120 synapses/group) are means ± SEM from 4 independent preparations. Statistical analysis was performed with two-way ANOVA followed by the Bonferroni's multiple comparison test. **$p<0.01$ versus control; °$p<0.05$, °°$p<0.01$, °°°$p<0.001$ versus the respective HRP-loaded sample.

The following figure supplement is available for figure 7:

**Figure supplement 1.** Morphometric analysis at the different time point in the HRP functional assay.

trafficking of SVs via intermediate endosomal compartments because of a defect in the direct recycling route bypassing intermediate endosomes. For Arf6-KD neurons, we favor the second hypothesis as morpho-functional experiments revealed that Arf6-depleted synapses show deficient SV formation upon stimulation but functional SV reformation from intraterminal Elos during the post-stimulus period. Endosome-like structures seem therefore to preferentially form at the expense of SVs in the absence of Arf6, which instead does not appear to play a major role on SV reformation from Elos.

Additional Arf6 downstream targets, other than AP-2, could also be involved, such as the phosphatidylinositolphosphate-kinase type Iγ, already described as a SV cycle regulator (*Di Paolo et al., 2004*), or the scaffold protein JIP3. Interestingly, JIP3 has been identified as an Arf6 target mediating the fast component of constitutive endocytosis (*Montagnac et al., 2011*) and the *C. elegans*

JIP3 homologue UNC-16 mutant shows endosomal structure accumulation and SV depletion at motor neuron synapses (*Brown et al., 2009*).

The hypothesis that SVs preferentially travel via endosomes when Arf6 expression/activation is impaired is also in accordance with the increased RRP, which was reported to go through endosomal sorting (*Hoopmann et al., 2010*). Indeed, RRP is increased at Arf6-depleted synapses but the kinetics of endocytosis for the SVs belonging to RRP was unaffected, supporting the idea that endosomal recycling is the retrieval mechanisms selected by SVs from the RRP. In line with these observations, clathrin depletion was recently reported to be ineffective on endocytotic decay after RRP depletion (*Kononenko et al., 2014*), suggesting a clathrin/Arf6-independent SV retrieval for RRP vesicles. However, the increased RRP due to Arf6 silencing was insensitive to TTX, opening the possibility that it is a distinct activity-independent mechanism. Differently from what we observed after the short (2 s) stimulation, Arf6 depletion impacts on SV endocytosis after the longer (20 s) stimulation, suggesting that not only the frequency, but also the duration of the stimulus can challenge the membrane retrieval mechanisms.

Whereas the Rab family of small GTPases has been extensively implicated in various steps of SV cycling, ranging from the docking/priming phases during exocytosis to recycling phases (*Binotti et al., 2015*; see for a review *Pavlos and Jahn, 2011*), our work reveals additional roles for the small GTPase of the Arf family, Arf6, in the SV cycling pathway and in the definition of the size of the SV pools. Interestingly, Rab proteins can act both as regulators or effectors for activated Arf6, as demonstrated in various cellular systems (*Chesneau et al., 2012*; *Shi et al., 2012*; *Pelletán et al., 2015*). In particular Arf6 and Rab35 have been shown to turn each other off by recruitment of reciprocal GAPs (*Chaineau et al., 2013*; *Dutta and Donaldson, 2015*) and active Rab35 is reported to regulate SV endosomal sorting (*Uytterhoeven et al., 2011*). Future work is needed to unravel the Arf6 interactome at the presynapse in order to define the specific GEFs and GAPs that regulate Arf6 activity in discrete subsynaptic compartments and the Arf6 effectors that mediate the described functions.

The use of pharmacological tools to mimic or revert Arf6 function at synapse opens new therapeutic approaches to correct dysregulations of the Arf6 pathway occurring in human neurological diseases associated with mutations in the Arf6 regulatory genes (*Shoubridge et al., 2010*; *Falace et al., 2010*; *Rauch et al., 2012*; *Fine et al., 2015*).

## Materials and methods

### Animals
Sprague-Dawley rats were from Charles River. All experiments were performed in accordance with the guidelines established by the European Community Council (Directive 2010/63/EU of March 4th 2014) and approved by the Italian Ministry of Health.

### Materials
Amino-5-phosphonopentanoic acid (D-APV), 6-cyano-7-nitroquinoxaline-2,3-dione (CNQX), tetrodotoxin (TTX), BafilomycinA1, and SecinH3 were from Tocris Bioscience (Cookson, USA). Restriction enzymes were from New England Biolabs (Ipswich, USA). Cell culture media were from Invitrogen (Carlsbad, CA, USA). Soluble horseradish peroxidase (HRP) type VI, 3,3′-Diaminobenzidine (DAB) ( #P6782 and #D8001) and all other chemicals were from Sigma-Aldrich, St. Louis, MO. The following primary antibodies have been used: anti-GFP (#MAB3580; Millipore, Darmstadt, Germany); anti-synaptobrevin-2 (Syb2; #104 202; Synaptic Systems, Goettingen, Germany); anti-actin (#A4700) and anti-Arf6 (#A5230, Sigma); anti-Arf6 (#ARP54598_P050; Aviva, San Diego, CA); anti-Rab5(#108 011 Synaptic Systems); anti-Arf1 (#05–1427; EMD Millipore); anti-V5 (#60708; Invitrogen); anti-β-tubulin (#T8328; Sigma) anti-p38 (Mab5258, EMD Millipore); anti-Homer (#106 011; Synaptic Systems); anti-Vglut1 (#135 304; Synaptic Systems).

### DNA cloning
Two shRNAs targeting the coding sequence (shRNA#1, CGCCAGGAGCTGCACCGCATTATCAAT-GA, shRNA#2, CGGATTCGGGACAGGAACTGGTATGTGCA) of *Rattus norvegicus* ADP-ribosylation factor 6 (Arf6) mRNA (GenBank accession no. NM_024152) cloned into pRFP-C-RS plasmid have

been acquired from OriGene Technologies (Rockville, MD). Non-effective mismatch shRNA was obtained by inserting four point mutations in the nucleotide sequence of shRNA#1 (CGCGAGCACC-TGCAGCGCATTATGAATGA). RNAi resistant Arf6 isoform, harboring nine silent point mutations into the rat *Arf6* mRNA sequence targeted by shRNA#1 (CGGCAAGAACTCCATCGGATAATTAA-CGA), has been provided by Biomatik (Cambridge, ON, Canada) and then subcloned into a pcDNA3.1/V5-His-Topo plasmid (Life Technologies, Carlsbad, CA) or into lentiviral vector pLVX-Ires-mCherry (Clontech, Mountain View, CA).

Oligonucleotides corresponding to shRNA#1 (Forward: 5'-AATT—CGCCAGGAGCTGCACCGCA-TTATCAATGA—CTCGAG—TCATTGATAATGCGGTGGCAGCTCCTGGCG—TTTTTTTAT-3; Reverse 5'-AAAAAAA— CGCCAGGAGCTGCACCGCATTATCAATGA —CTCGAG— TCATTGATAATGCGG-TGGCAGCTCCTGGCG -3') and the mismatch control (Forward: 5'-AATT—CGC**G**AG**C**AC**C**TG-CA**G**CGCATTAT**G**AATGA—CTCGAG—TCATTCATAATGCGCTGCAGGTGCTCGCG—TTTTTTTAT-3'; Reverse: 5'-AAAAAAA— CGCGAGCACCTGCAGCGCATTATGAATGA —CTCGAG— TCATTCA-TAATGCGCTGCAGGTGCTCGCG -3') were inserted into the lentiviral vector pLKO.3G (Addgene, Cambridge, MA) using *Pac*I and *Eco*RI restriction sites. Insertions were confirmed by single digestion with *Psh*AI followed by direct Sanger sequencing. The production of vesicular stomatitis virus-pseudotyped third-generation lentiviruses was performed as described previously (*Verstegen et al., 2014*). Viral titers ranging from 1.0 to 8.0 x $10^8$ transforming units/ml were obtained.

## Hippocampal primary culture

Pregnant rats were killed by inhalation of $CO_2$, and 18-day-old embryos were immediately removed by Cesarean section. Hippocampi were dissected and dissociated by enzymatic digestion in 0.125% trypsin for 20 min at 37°C and then triturated with a fire-polished Pasteur pipette. No antimitotic drugs were added to prevent glia proliferation. Neurons were plated on poly-L-lysine (0.1 mg/ml; Sigma-Aldrich)-treated 25-mm glass coverslips at a density of 40,000–60,000 cells per coverslip and on poly-L lysine (0.01 mg/ml)-treated 35-mm plastic wells at a density of 300,000 cells per well for immunoblots.

## Cells transfection and transduction

HeLa cells were transfected with Lipofectamine 2000 reagent (Life Technologies) according to manufacturer's instructions and processed after 48 hr for immunoblotting. Primary hippocampal neurons were nucleofected at 0 DIV with 3 µg of sh-Arf6-RFP or mismatched-Arf6-RFP constructs with AMAXA P3 primary cell 4D-Nucleofector kit (Lonza) and processed for western blotting at 7 DIV. For live imaging experiments neurons were transfected at 12–14 DIV with 0.5 µg of sh-Arf6-RFP or mismatched-Arf6-RFP, resistant Arf6-V5 and SypHy constructs using Lipofectamine 2000 and analyzed 3–5 days post-transfection. Primary hippocampal neurons were infected with lentiviruses for sh-Arf6 and respective mismatched version at 12 DIV at 10 multiplicity of infection (MOI). After 20 h, half of the medium was replaced with fresh medium and EM experiments were performed between 17 and 18 DIV.

## Immunoblotting and immunocytochemistry

Primary hippocampal neurons were fixed with 4% paraformaldehyde in PBS for 20 min, permeabilized with 0.1% Triton X-100 in PBS for 2 min, blocked 30 min with 5% BSA in PBS before incubation with primary antibodies, followed by Alexa Fluor-conjugated 405, 488, 568 and 647 secondary antibodies. For immunoblotting, neurons or HeLa cells were scraped in lysis buffer (150 mM NaCl, 50 mM Tris, NP-40 1%, SDS 0.1%) plus protease and phosphatase inhibitors (0.2 mM PMSF, 2 µg/ml pepstatin, 1 mM NaF, and 1 mM $NaVO_4$). After centrifugation at 16,000 x *g* for 10 min at 4°C, samples were loaded on SDS-PAGE gel, transferred to nitrocellulose membranes, and immunoblotted with primary antibodies followed by HRP-conjugated secondary antibodies.

## Synaptosome preparation and fractionation

Synaptosomes were purified from the P2 fractions by centrifugation on discontinuous Percoll gradient from the pooled cortex of two adult mice (3–6 months old). The tissue was homogenized in 10 ml of ice-cold HB buffer (0.32 M sucrose, 1 mM EDTA, 10 mM Tris, pH 7.4) containing protease inhibitors, using a glass-Teflon homogenizer. The resultant homogenate was centrifuged at 1000 ×

g for 5 min at 4°C in order to remove nuclei and debris and the supernatant was further centrifuged at 18,900 × g for 10 min at 4°C to obtain the P2 fraction. The pellet was resuspended in 2 ml of HB buffer and gently stratified on a discontinuous Percoll gradient (3%–10%–23%). After a 10 min centrifugation at 18,900 × g at 4°C, the synaptosomal fraction from the 10% and 23% Percoll interface was collected, washed in Krebs buffer (140 mM NaCl, 5 mM KCl, 5 mM NaHCO$_3$, 1.3 mM MgSO$_4$, 1 mM phosphate buffer pH 7.4, 10 mM Tris/Hepes pH 7.4) to eliminate Percoll and used as the starting material for subsequent ultrafractionation. Ultrasynaptic fractionation was performed as previously described (*Phillips et al., 2001*). Briefly, synaptosomes were pelleted by centrifugation (16,000 × g, 5 min, 4°C) and resuspended in 300 µl of 0.32 M sucrose, 0.1 mM CaCl$_2$. An aliquot was removed and kept as total. Protease inhibitors were used in all purification steps. Synaptosomes were then diluted 1:10 in ice-cold 0.1 mM CaCl$_2$ and mixed with an equal volume of 2X solubilization buffer (2% Triton X-100, 40 mM Tris, pH 6, 4°C). After a 30 min incubation at 4°C, the insoluble material (synaptic junction) was pelleted by centrifugation (40,000 × g, 30 min, 4°C). The supernatant (1% TX-100 pH 6 soluble) was decanted and the proteins precipitated with 6 volumes of acetone at -20°C overnight and then centrifuged (18,000 × g, 30 min, -15°C). The synaptic junction pellet (containing the insoluble postsynaptic density and the presynaptic active zone) was resuspended in 10 volumes of 1X solubilization buffer (1% Triton X-100, 20 mM Tris, pH 8, 4°C), incubated for 30 min at 4°C and then centrifuged (40,000 × g, 30 min, 4°C). The proteins in the supernatant (1% TX-100 pH 8 soluble) were precipitated in 6 volumes of acetone at -20°C overnight and then centrifuged (18,000 × g, 30 min, -15°C), while the pellet was kept as postsynaptic density fraction (1% TX-100 pH 8 insoluble). All pellets were resuspended in 5% SDS, quantified by BCA assay, and loaded on SDS-PAGE gels for electrophoresis and consecutive western blotting.

## Arf6-activation assay

Arf6 activation was determined using an Arf6-GTP-specific pull-down assay (Cytoskeleton, Denver, CO, USA), according to manufacturer instructions. Samples were fractionated by SDS-12% PAGE and Arf6-GTP and total Arf6 level were evaluated by western blotting using an anti-Arf6 antibody (Sigma Aldrich).

## Electron microscopy

Primary hippocampal neurons were fixed with 1.2% glutaraldehyde in 66mM sodium cacodylate buffer, pH 7.4, postfixed in 1% OsO$_4$, 1.5% K$_4$Fe(CN)$_6$, and 0.1M sodium cacodylate, *en bloc* stained with 1% uranyl acetate, dehydrated, and flat embedded in epoxy resin (Epon 812, TAAB). Ultrathin sections (70 nm) were collected on copper mesh grids (EMS) and observed with a Jeol JEM-1011 microscope at 100 kV equipped with an ORIUS SC1000 CCD camera (Gatan, Pleasanton, CA). Morphometric analysis was done using NIH ImageJ. Structures with sagittal diameter comprised between 20 and 80 nm were classified as SVs, while those with a sagittal diameter bigger than 80nm were classified as intra-terminal cisternae. SVs touching AZ were classified as docked SVs.

## 3D reconstruction

For 3D reconstructions, 60-nm serial sections were collected on carbon-coated copper slot formwar grids and serial synaptic profiles were acquired as described for mono-dimensional TEM. Serial electron micrographs were aligned with Midas of IMOD software (Colorado University, CO, USA).

## Live imaging experiments

Primary hippocampal neurons were maintained in Tyrode's solution (140 mM NaCl, 3 mM KCl, 2 mM CaCl$_2$, 1 mM MgCl$_2$, 10 mM HEPES-buffered to pH 7.4, 10 mM glucose, CNQX 10 µM; APV 50 µM) through a laminar flow perfusion system. For electrical field stimulation, coverslips were mounted into the imaging chamber (~100 µl volume; Quick Exchange Platform; Warner Instruments) and APs were evoked by passing 1 msec current pulses through platinum iridium electrodes using an AM2100 stimulator (AM-Systems). After 10 s of baseline acquisition (*F*0), neurons were stimulated with a train of 40/400 APs at 20Hz in the presence or absence of 1 µM Bafilomycin. At the end of the stimulation protocol, cells were perfused with 50 mM NH$_4$Cl ($F_{max}$). Images were analyzed using the eXcellence software (Olympus). The total increase in the fluorescence signal ($\Delta F$) was calculated by subtracting *F*0 and the $\Delta F$ was normalized to the fluorescence value obtained by alkalization of

the entire vesicle pool using $NH_4Cl$ ($\Delta F_{max}$). The time constant of endocytosis ($\tau$ endo) was calculated by the fitting the post-stimulus decay with a single-exponential function. To quantify net exocytosis (EXO) ongoing during stimulation, the areas below the curves obtained in the presence of Bafilomycin were measured. To quantify the net during stimulus' endocytosis (ENDO), additional curves were built from the differences between the curves run in the presence and those in the absence of Bafilomycin and the areas below the curves were measured.

To evaluate the SypHy surface fraction, neurons were perfused for 20 sec with Tyrode Solution (pH 7.4) to measure the basal fluorescence ($F_{basal}$), then with MES buffer (pH 5.5) ($F_{acid}$) to measure the fluorescence under the acidic condition, and lastly with a Tyrode solution with $NH_4Cl$ ($F_{alkalin}$) to evaluate the total amount of probes at the synaptic boutons. SypHy surface fraction was evaluated by subtracting the fluorescence change between basal and acidic conditions ($F_{basal}$-$F_{acid}$) from the fluorescence change between alkaline and acidic conditions ($F_{alkalin}$-$F_{acid}$) and reported as percent values, as follows: $F_{extra} = [(F_{basal}-F_{acid})/ (F_{alkalin}-F_{acid})]*100$.

Data are expressed as means $\pm$ SEM for the number of coverslips analyzed. The number of synapses analyzed per coverslip ranged between 20 and 40. Three to five cells preparation were performed for each experimental condition.

### HRP functional assay

Primary hippocampal neurons were stimulated as above in presence of 10mg/ml soluble HRP. At the end of the stimulus, soluble HRP was washed out and neurons were fixed either immediately or after 2 or 20 min. After chemical fixation, neurons were washed in 0.1 M cacodylate buffer and incubated for 10 min

in a solution containing 0.3 mg/ml of 3,3-Diaminobenzine (DAB) in 0.1 M cacodylate buffer. Neurons were incubated in a solution containing 0.3 mg/ml of DAB + 0.003% $H_2O_2$ in 0.1 M cacodylate buffer until a brown substrate developed, to allow for HRP peroxidation. Neurons were then rinsed in cold distilled water, to block DAB peroxidation and post-fixed in 1% $OsO_4$, 1.5% $K_4Fe(CN)_6$, 0.1 M sodium cacodylate. The numbers of HRP-positive SVs and cisternae were evaluated using NIH ImageJ.

### Statistical analysis

Data were analyzed using the Student's *t*-test or, in case of more than two experimental groups, by one- or two-way ANOVA, followed by *post hoc* multiple-comparison tests (Bonferroni's or Dunnet's) using GraphPad Prism or SigmaPlot software. The significance level was set at p<0.05.

## Acknowledgements

We thank Elisa Belmonte and Nicola Vanni for help and advice and Silvia Casagrande for technical assistance. Yongling Zhu and Charles F. Stevens (The Salk Institute, La Jolla, CA, USA) for the synaptophysin-pHluorin construct. Support for this work was given by Jerome Lejeune Foundation and Università degli Studi di Genova to AF. EU FP7 Integrating Project 'Desire' (Grant no. 602531), to FB and AF is also acknowledged.

## Additional information

### Funding

| Funder | Grant reference number | Author |
|---|---|---|
| Fondation Jérôme Lejeune | 1155-FA2013A | Anna Fassio |
| Università degli Studi di Genova | D34G1300017005 | Anna Fassio |

The funders had no role in study design, data collection and interpretation, or the decision to submit the work for publication.

## Author contributions
ET, MF, Acquisition of data, Analysis and interpretation of data; AFal, Drafting or revising the article, Contributed unpublished essential data or reagents; FB, Conception and design, Drafting or revising the article; AFas, Conception and design, Analysis and interpretation of data, Drafting or revising the article

## Ethics
Animal experimentation: All experiments were carried out in accordance with the guidelines established by the European Communities Council (Directive 2010/63/EU of March 4th 2014) and were approved by the Italian Ministry of Health.

## Additional files

**Supplementary files**
• Supplementary file 1. Synaptic area and AZ length in the various experimental groups.

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
