## [Decision Letter]

Thank you for submitting your work entitled "Arf6 acts as a presynaptic chaperone in determining direct versus endosomal recycling of synaptic vesicles" for peer review at *eLife*. Your submission has been evaluated by a Senior editor and three reviewers, one of whom is a member of our Board of Reviewing Editors.

The reviewers have discussed the reviews with one another and the Reviewing editor has drafted this decision. While your work is largely seen as novel and interesting, the reviewers have some concerns.

Summary:

This study examines the role of Arf6 inactivation on synaptic vesicle cycling at presynaptic nerve terminals. This paper is significant as it provides important insights in presynaptic endosome function, an area that is so far been poorly studied due to lack of tools and low depth of analysis.

As most of the data appear solid, and the experiments are well performed, some experimental controls are needed to improve the rigidity of the conclusions drawn. Also the interpretation of the data should be carefully revisited.

Essential revisions:

Control experiments:

1) The authors need to provide better data to convince the presence of Arf6 in the pre synapse. While this is likely due to the observed phenotypes, a more direct demonstration may be helpful.

Related, while the labeling of Figure 2 shows that disruption of Arf6 alters endosomal markers, this is mainly through modification of postsynaptic Arf6 signaling. Furthermore, the conclusion that Arf6 activity, rather than the protein, mediates the phenotype would be stronger if the authors can assess if Arf6 protein was still present at synapses with the inhibitor present. Maybe the drug leads to delocalization of Arf6.

2) Showing the ultrastructure of the Arf6 rescue would strengthen the paper.

Please also comment on and revise these points:

The authors suggest that Arf6 is a "switch channeling SVs to direct rather than to endosomal recycling" (highlights). Figure 5 implies that this statement is false. The absence of Arf6 actually implies that more vesicles can be immediately recycled upon stimulation, when the authors use a stimulation protocol that is well within the physiological range of these cultures.

The authors suggest that Arf6 "controls the availability of SVs for release" (highlights). Figure 5 and Figure 6 imply that this statement is also false. Plenty of vesicles are available for release in the absence of Arf6 – in fact, about as many as in control synapses, even with long stimuli (Figure 6).

The authors suggest that Arf6 has an unexpected role in "determining the size of the readily releasable pool of SVs". This is interpreted from observing a larger size of this pool in the absence of Arf6, which implies that Arf6 may, if anything, have a role in reducing this pool. This statement is therefore confusing.

The authors' data demonstrate that the Arf6 absence slows endocytosis (Figure 6), which results in the appearance of large endosome-like objects. It is evident that these objects do participate in synaptic recycling. This can be inferred from the fact that they disappear upon TTX treatment, and also from the results in Figure 7, which follow closely the original Heuser & Reese, 1973, experiments on this subject. The simplest interpretation of the manuscript, therefore, is that Arf6 is involved in the break-up of such organelles in synaptic vesicles. In a knock-down of Arf6 the efficiency of this break-up process is far lower, and the endosome-like organelles become prominent, since their break-up is slowed down. Other effects, such as the persistence of the readily recycling pool, point to a lower involvement of endosome-like organelles in this pool, something that has been suggested before, as the authors also acknowledge.

In the Discussion and the title the authors talk about Arf6 acting as a chaperone. There is no evidence in the paper that Arf6 acts as a chaperone so we would advise you to remove this statement.

---

## [Author Response]

*Control experiments:*

*1) The authors need to provide better data to convince the presence of Arf6 in the pre synapse. While this is likely due to the observed phenotypes, a more direct demonstration may be helpful.*

Related, while the labeling of Figure 2 shows that disruption of Arf6 alters endosomal markers, this is mainly through modification of postsynaptic Arf6 signaling.

Following this suggestion we have performed immunocytochemistry on rat hippocampal cultures at 17-18 DIV (the very same time window of our structural and functional experiments) with anti-Arf6 antibodies (AVIVA SYSTEM BIOLOGY, ARP54598-P050, 1:100) and either presynaptic (Synaptophysin, Millipore, Mab5258, 1:500) or postsynaptic (Homer1, Synaptic System 160 011, 1:300) markers. As shown by the representative images and related intensity profiles (panel A of new Figure 1—figure supplement 1) Arf6 signal colocalized with both markers. Considering the resolution of conventional optical microscopy this evidence is only suggestive for expression at both pre- and post-synaptic compartments. We therefore performed immunocytochemistry with Arf6, the presynaptic marker VGlut1 and the postsynaptic marker Homer1. We used VGlut1 because an antibody raised in guinea pig anti-VGLut1 is commercially available (Synaptic Systems, #135 304, 1:200) allowing to perform triple immunolabelling. We analysed the intensity profiles of the Arf6 signal at single synaptic puncta in which pre- and post-synaptic signals were not completely overlapped. We revealed that indeed Ar6 signal is present at both pre- and post-synaptic compartments (panel B of new Figure 1—figure supplement 1). This evidence is supported by biochemical fractionation of isolated nerve terminals in which Arf6 immunoreactivity is mainly associated with intra-synaptic protein, as the SV-marker Synaptophysin, and scarcely associated with both presynaptic- (active zone) and post-synaptic (postsynaptic density) membranes (panel C of new Figure 1—figure supplement 1). It is therefore expected that, as correctly observed in the revision, disruption of Arf6 may also alter endosomal markers at the postsynaptic level.

*Furthermore, the conclusion that Arf6 activity, rather than the protein, mediates the phenotype would be stronger if the authors can assess if Arf6 protein was still present at synapses with the inhibitor present. Maybe the drug leads to delocalization of Arf6.*

We thank the reviewers for the suggestion and performed synaptic localization experiments in control (DMSO 3hrs) and sSecinH3-treated (30μM 3 hrs) cultures and revealed no significant dispersion of the synaptic Arf6 signal upon treatment. These results have been inserted in panel B of new Figure 3.

*2) Showing the ultrastructure of the Arf6 rescue would strengthen the paper.*

Rescue of the ultrastructural phenotype has been added as supplementary to Figure 1 (new Figure 1—figure supplement 3). The quantification of the synaptic area and AZ length of these new samples have been added to Table 1.

*Please also comment on and revise these points:*

*The authors suggest that Arf6 is a "switch channeling SVs to direct rather than to endosomal recycling" (highlights). Figure 5 implies that this statement is false. The absence of Arf6 actually implies that more vesicles can be immediately recycled upon stimulation, when the authors use a stimulation protocol that is well within the physiological range of these cultures.*

As pointed out in the above comment, the stimulation protocol applied in the experiments reported in Figure 5 (40 action potentials at 20Hz) is well within physiological range of the cultures. However, under our experimental conditions recycling is rather small (if any) during this stimulation protocol, as demonstrated by the similar peak response in the presence or absence of Bafilomycin (see Figure 5). Thus, the results of Figure 5 imply that the availability of SV for release is increased in the absence of Arf6, in line with the ultrastructural data showing that the number of docked vesicles is also increased (see Figure 1). The highlight "Arf6 switch channeling SVs to direct rather than to endosomal recycling" was therefore not related to results in Figure 5 but rather to the results shown in Figure 6 and Figure 7, in which a different protocol of stimulation (400 action potentials at 20Hz) has been used. This protocol allows measurable recycling that was indeed significantly decreased in the absence of Arf6. For the sake of clarity we modified the statement in the highlight with: “Arf6 regulates SV recycling”.

*The authors suggest that Arf6 "controls the availability of SVs for release" (highlights). Figure 5 and Figure 6 imply that this statement is also false. Plenty of vesicles are available for release in the absence of Arf6 – in fact, about as many as in control synapses, even with long stimuli (Figure 6).*

We perfectly agree with the referees in saying that plenty of SVs are available for release in the absence of Arf6 and modified the highlight into “Arf6 defines the readily releasable pool of SVs”.

*The authors suggest that Arf6 has an unexpected role in "determining the size of the readily releasable pool of SVs". This is interpreted from observing a larger size of this pool in the absence of Arf6, which implies that Arf6 may, if anything, have a role in reducing this pool. This statement is therefore confusing.*

We agree with the referees that active Arf6 reduces the readily releasable pool of SVs, as unmasked by the increased RRP in the absence of Arf6. By using “determining” we meant that Arf6 define the readily releasable pool of SVs (see novel highlight) as demonstrated by the increased size of the pool in its absence.

*The authors' data demonstrate that the Arf6 absence slows endocytosis (Figure 6), which results in the appearance of large endosome-like objects. It is evident that these objects do participate in synaptic recycling. This can be inferred from the fact that they disappear upon TTX treatment, and also from the results in Figure 7, which follow closely the original Heuser & Reese, 1973, experiments on this subject. The simplest interpretation of the manuscript, therefore, is that Arf6 is involved in the break-up of such organelles in synaptic vesicles. In a knock-down of Arf6 the efficiency of this break-up process is far lower, and the endosome-like organelles become prominent, since their break-up is slowed down. Other effects, such as the persistence of the readily recycling pool, point to a lower involvement of endosome-like organelles in this pool, something that has been suggested before, as the authors also acknowledge.*

We agree with the referees that the simplest interpretation of the increased endosome-like organelles at Arf6-depleted synapses would be that Arf6 is involved in the break-up of such organelles in synaptic vesicles, which therefore persist in the terminal when Arf6 function is impaired. To validate this hypothesis we performed morpho-functional experiments shown in Figure 7, in which the ultrastructural analysis was coupled with the evaluation of SV and organelle recycling upon stimulation. The data suggest that endosome-like organelles preferentially form at the expense of SVs upon Arf6 depletion. Indeed, the increased endosome-like structures upon stimulation (400 action potentials at 20Hz) at Arf6-depleted synapses is followed by formation of HRP-positive SVs, as shown in the quantification after 2 min. of washing, which most probably derive from HRP-positive endosome-like organelles. Moreover after longer washing (20 min) HRP-positive-SVs and HRP-positive endosome-like structures equally decreased in both control and Arf6-depleted synapses. These data therefore suggested a preferential accumulation rather than an increased persistence of endosomal-like organelles at Arf6 depleted synapses.

*In the Discussion and the title the authors talk about Arf6 acting as a chaperone. There is no evidence in the paper that Arf6 acts as a chaperone and this is better removed.*

The term “chaperone” has been removed from both title and manuscript text.